# Serum Asymmetric Dimethylarginine Levels in Patients with Vasovagal Syncope

**DOI:** 10.3390/medicina55110718

**Published:** 2019-10-29

**Authors:** Adem Atıcı, Gonul Aciksari, Omer Faruk Baycan, Hasan Ali Barman, Mehmet Rasih Sonsöz, Mustafa Sahin, Ramazan Asoglu, Ahmet Demirkıran, Şeref Kul, Baris Gungor, Eser Durmaz, Ahmet Kaya Bilge, Irfan Sahin

**Affiliations:** 1Cardiology Department, Istanbul Medeniyet University, Goztepe Training and Research Hospital, 34722 Istanbul, Turkey; drgonulkutlu@hotmail.com (G.A.); omer.baycan@medeniyet.edu.tr (O.F.B.); doktorseref@hotmail.com (Ş.K.); 2Cardiology Department, Okmeydani Training and Research Hospital, 34384 Istanbul, Turkey; drhasanali@hotmail.com; 3Cardiology Department, Istanbul University Istanbul School of Medicine, 34093 Istanbul, Turkey; mrsonsoz@gmail.com (M.R.S.); aademirkiran@gmail.com (A.D.); ahmetkayabilge@hotmail.com (A.K.B.); 4Biochemistry Department, Hitit University, Erol Olcok Training and Research Hospital, 19040 Corum, Turkey; mustafasahinist@gmail.com; 5Cardiology Department, Adiyaman Training and Research Hospital, 02200 Adiyaman, Turkey; dr.asoglu@yahoo.com; 6Department of Cardiology, Siyami Ersek Cardiothoracic Surgery Center, 34668 Istanbul, Turkey; drbarisgungor@gmail.com; 7Cardiology Department, Istanbul University-Cerrahpasa, Cerrahpasa School of Medicine, 34096 Istanbul, Turkey; durmazeser@hotmail.com; 8Cardiology Department, Bagcilar Training ve Research Hospital, Bagcilar Center, 34100 Istanbul, Turkey; dr.irfansahin@gmail.com

**Keywords:** Vasovagal syncope, head-up tilt test, asymmetric dimethylarginine, nitric oxide

## Abstract

*Background and Objectives*: Vasovagal syncope (VVS) is the most common cause of syncope and has multiple pathophysiological mechanisms. Asymmetric dimethylarginine (ADMA) is the major inhibitor of nitric oxide (NO). In this study, we aimed to investigate the relationship between plasma ADMA levels and syncope during the head-up tilt (HUT) test. *Materials and Methods*: Overall, 97 patients were included in this study. They were above 18 years of age and were admitted to our clinic with the complaint of at least one episode of syncope consistent with VVS. The HUT test was performed in all patients. Patients were divided into the following two groups based on the HUT test results: group 1 included 57 patients with a positive HUT test and group 2 included 35 patients with a negative HUT test. Blood samples were taken before and immediately after the HUT test to measure ADMA levels. *Results*: No significant intergroup differences were observed concerning gender and age (female gender 68% vs 60%; mean age 24.85 ± 4.01 vs 25.62 ± 3.54 years, respectively, for groups 1 and 2). ADMA values were similar between groups 1 and 2 before the HUT test [ADMA of 958 (544–1418) vs 951 (519–1269); *p* = 0.794]. In the negative HUT group, no significant differences were observed in ADMA levels before and after the HUT test [ADMA of 951 (519–1269) vs 951 (519–1566); *p* = 0.764]. However, in the positive HUT group, ADMA levels were significantly decreased following the HUT test [pretest ADMA of 958 (544–1418) vs post-test ADMA of 115 (67–198); *p* < 0.001]. *Conclusion*: ADMA levels significantly decreased after the HUT test in patients with VVS.

## 1. Introduction

Syncope is defined as the transient loss of consciousness, loss of muscle tone, and inability to sustain activity because of cerebral hypoperfusion, which reverses spontaneously [1]. Vasovagal syncope (VVS) is the most common subtype of syncope and has multiple pathophysiologic mechanisms. Despite a high morbidity rate, VVS has a benign prognosis [2,3,4]. Although the exact pathophysiologic mechanism of VVS is unclear, it has been previously demonstrated that neurohumoral activation increases the plasma levels of several vasoactive substrates, such as vasopressin, adrenalin, nitric oxide (NO), endothelin, and cortisol during the head-up tilt (HUT) test [5,6]. NO exerts a powerful vasodilatory effect and plays a critical role in patients with VVS [7]. Orthostatic intolerance during psychological stress, long-term exposure to gravity, and bleeding are implications of this condition [8,9,10]. Moreover, excessive NO production adversely affects the autonomic nervous system by increasing the vagal tone and producing a sympatholytic effect [11,12,13]. Furthermore, it has been demonstrated that exogenous NO synthase (NOS) inhibitors prevent the hypotensive effects of excessive NO production on cardiovascular tone [14].

NO is synthesized from L-arginine by NOS and released from the endothelial cells in response to acetylcholine [15,16,17]. Of the two major inhibitors of NOS (Figure 1), asymmetric dimethylarginine (ADMA) is the most powerful endogenous inhibitor [18,19]. It has been previously demonstrated that neurohormonal mediators, such as ADMA, exert their effects on the vascular system through NO [20]. Recent experimental and clinical trials have revealed that even small variations in plasma ADMA levels alter NO synthesis, vascular tone, and systemic vascular resistance [21]. Moreover, ADMA decreases endothelium-dependent vasodilation and renal sodium excretion, and increases blood pressure through NOS inhibition [22].

To the best of our knowledge, the current literature has no study that has investigated the role of ADMA in VVS. Hence, in this study, we aimed to investigate the relationship between plasma ADMA levels and syncope during the HUT test in healthy individuals admitted with a complaint of syncope.

## 2. Materials and Methods

We prospectively included 97 patients who were admitted to our clinic between 2017 and 2019 with a complaint of at least one episode of syncope consistent with VVS. After the patients gave their informed consent, they were included in this study in accordance with the Declaration of Helsinki. Full ethical approval for this study was obtained from the Ethics Committee of the Istanbul University, Turkey (2017/944), approved on 29 June 2017. The exclusion criteria were as follows: (1) any kind of structural heart disease; (2) a previous diagnosis of metabolic disease, orthostatic hypotension, chronic kidney disease, or chronic neurologic disorders; and (3) any kind of conduction disorders, including first-degree atrioventricular block. Five patients who were thought to be conversion cases were excluded from the study. Baseline demographics, clinical characteristics, and 12-lead electrocardiograms of the participants were recorded.

Transthoracic echocardiography was performed before the HUT test by using a Philips IE33 device with an X5-1 transthoracic probe. A standard echocardiographic evaluation was carried out using M-mode, two-dimensional, and Doppler studies per the guidelines of the American Society of Echocardiography. Each patient underwent the HUT test. The diagnosis of VVS was made per the guidelines of the European Society of Cardiology [23]. Patients were excluded from the study in the event of fainting or any other kind of complaint without altered heart rate (HR) or blood pressure.

### 2.1. The HUT Test Protocol

The HUT test was performed using an electronically controlled table which had a footboard capable of bearing the patient’s weight. Systolic and diastolic blood pressures, pulse rate, heart rhythm, and right forearm blood flow were closely monitored and recorded. Blood pressure was measured every minute using an automated system. The patients were kept on the tilt table in a supine position for 15 min after blood collection. Patients were tilted at 70° for up to 45 min without drug provocation [24,25]. If presyncope with hypotension occurred during the test, the tilt table was rapidly lowered to return the patient to the supine position, and the study was terminated. Bradycardia was defined as a 20% decrease from the baseline HR or a decrease of more than 20 beats in a minute. Hypotension was defined as a decrease in systolic blood pressure (SBP) of more than 20% from baseline or maximum blood pressure or more than 30 mmHg decrease of systolic pressure. Syncope of patients with a positive HUT test was classified based on the modified Vasovagal Syncope International Study (VASIS) classification as VASIS 1 (mixed), VASIS 2A (cardioinhibition without asystole), VASIS 2B (cardioinhibition with asystole), and VASIS 3 (vasodepressive) [26].

### 2.2. Blood Collection and Performance Characteristics of the ADMA Assay

Venous blood samples were collected from each patient before (15 min before test) and after (15 min after test) the HUT test to measure ADMA levels in patients with a diagnosis of VVS. Plasma levels of ADMA were assessed 15 min before the HUT test and 15 min after the test termination in case of test positivity or patient’s complaints. ADMA (Human ADMA enzyme-linked immunosorbent assay (ELISA), Wuhan Fine Biotech Co. Ltd, Hubei, China, Catalog no: EU2562) levels were determined using ELISA. Quality control was assured using two levels of material for each test. The detection range was 15.625–1000 ng/mL for ADMA. The reported intra-assay and inter-assay coefficient of variations were 6.6% and 6.9%, respectively.

### 2.3. Statistical Analysis

Statistical analyses were performed using the SPSS v.15.0 (SPSS Inc., Chicago, IL, USA) software. All continuous variables were presented as mean ± standard deviation or median (25th and 75th percentiles), regardless of normal or non-normal distribution of data. The distribution of continuous variables was evaluated using the Kolmogorov–Smirnov or Shapiro–Wilk test. Normally distributed data of the independent groups were compared using the independent samples student’s t-test. The Mann–Whitney U test was used if the data were non-normally distributed. The paired t-test or the Wilcoxon test—according to the distribution of the variables—was performed to evaluate the quantitative data before and after the HUT test. The relationships among parameters were assessed using Pearson’s or Spearman’s correlation analysis according to the normality of the data. Chi-square or Fisher’s exact test was used to determine the intergroup differences in categorical variables. All p values were two-sided in the tests, and *p* values less than 0.05 were considered statistically significant.

## 3. Results

The clinical characteristics and demographic variables of the study population are shown in Table 1. Patients were divided into the following two groups based on the HUT test results: 57 patients with a positive HUT test (group 1) and 35 patients with a negative HUT test (group 2). Although most patients were females in both groups (68% of group 1 and 60% of group 2), no significant intergroup differences were observed concerning gender and age. The mean age of participants in groups 1 and 2 was 24.85 ± 4.01 and 25.62 ± 3.54 years, respectively. No significant intergroup differences were observed regarding left ventricular ejection fraction (group 1: 65.10 ± 5.08%; group 2: 67.00 ± 5.71%; *p* = 0.101). HR and systolic and diastolic blood pressures before and after the HUT test were similar between groups (SBP 111.28 ± 11.37 mmHg vs 114.57 ± 6.68 mmHg, *p* = 0.124; DBP 69.87 ± 8.49 mmHg vs 71.28 ± 5.46 mmHg, *p* = 0.335; HR 73.56 ± 3.63 beats/min vs 74.62 ± 2.98 beats/min, *p* = 0.148, all respectively for groups 1 and 2). In addition, of the 57 patients with positive HUT, 30 were diagnosed with vasodepressor type syncope, and the remaining were diagnosed with mixed type.

ADMA values were similar between group 1 and group 2 before the HUT test [ADMA of 958 (544–1418) vs 951 (519–1269), respectively for groups 1 and 2; *p* = 0.794]. The negative HUT group exhibited no significant differences in ADMA levels before and after the HUT test [ADMA of 951 (519–1269) before the HUT test vs 951 (519–1566) after the test; *p* = 0.764]. However, in the positive HUT group, ADMA levels were significantly decreased following the HUT test [ADMA of 958 (544–1418) before the test vs 115 (67–198) after the test; *p* < 0.001]. Therefore, the positive HUT group had decreased ADMA levels compared with their pretest level, whereas no change in ADMA levels were observed for the negative HUT group (Table 2) (Figure 2).

Further analyses were conducted to compare the subgroups of positive HUT, and no significant difference was detected following the HUT test [ADMA of 123 (68–223) in the vasodepressor type VVS vs 99 (60–164) in the mixed type VVS; *p* = 0.187].

## 4. Discussion

In this study, we investigated the relationship between ADMA levels and syncope during the HUT test in patients with a suspicion of VVS. The principal findings of our study were as follows: (1) the ADMA levels of positive and negative HUT groups were similar; (2) patients with negative HUT had identical pre- and post-test ADMA values, whereas patients with positive HUT had decreased post-test ADMA levels compared with their pretest levels; and (3) subgroup analyses of the positive HUT group regarding vasodepressor or mixed type syncope revealed that their pre- and post-test ADMA values were similar.

Despite the current literature revealing several studies that have investigated VVS, the exact pathophysiology of VVS is still unclear. It has been suggested that the underlying mechanism of VVS involves alterations in the plasma levels of vasoactive substances (e.g., catecholamines, NO, endothelin, vasopressin, or prostanoids) in response to orthostatic stress [5,6]. Several studies have demonstrated that endothelium plays a crucial role in the regulation of vascular tone in VVS and endothelial hyperactivity through upregulation of NO synthase and is responsible for profound vasodilation [27,28]. The facts regarding nitrate hypersensitivity in patients with VVS and increased NO levels in patients with a positive HUT test supports the theory that NO has a crucial role in the underlying pathophysiologic mechanism of VVS [7,29,30]. Notably, the two forms of NO are neuronal and endothelial. Endothelial NO acts directly on the vasculature and causes vasodilation, whereas neuronal NO acts indirectly via the autonomic nervous system. Neuronal NO increases parasympathetic activation and decreases sympathetic activation, which eventually leads to the augmented synaptic transmission of acetylcholine with resultant vasodilation [31,32,33].

The major inhibitor of NO biosynthesis in humans is ADMA [34]. The effect of NO biosynthesis inhibition on the vascular bed is characterized by a shift in endothelial hemostasis toward vasoconstriction and endothelial dysfunction [35]. Exogenous ADMA infusion has been previously shown to increase systemic vascular resistance, mean arterial pressure, and pulmonary vascular resistance (in men) and decrease HR and cardiac output [36]. Moreover, when ADMA levels decreased, NO synthesis increased, and arterial blood pressure and systemic vascular resistance decreased [37]. Therefore, we investigated the changes in ADMA values before and after the HUT test in patients with a suspicion of VVS.

Some experimental data suggest that NO might play a role in the vasodilation occurring during the HUT test. The study of Shi Y et al. involving a small number of patients with a diagnosis of VVS demonstrated that NO levels were significantly higher in patients with a positive HUT test compared with those with a negative HUT test [7]. Our study demonstrated that the levels of plasma ADMA—a NOS inhibitor—were significantly lower during syncope triggered by the HUT test. Based on both findings, it is logical to assume that an increase in the NO level, which is believed to play a major role in the pathophysiology of syncope, might be due to a decrease in the NOS inhibitor level. Furthermore, Dietz et al. have demonstrated that exogenous N-monomethyl-L-arginine acetate (L-NMMA) infusion, which is a NOS inhibitor, decreased the NO levels in patients with VVS. Stewart et al. also measured the effect on NO through an inhibitor of NOS (Nc-monomethyl-L-arginine) and demonstrated that the exogenous L-NMMA infusion, which also decreases NO synthesis indirectly, improved the blood pressure in patients with VVS and had beneficial effects on orthostatic intolerance. Their data demonstrated that impaired splanchnic adrenergic vasoconstriction in young patients with VVS, the most common form of syncope, accounts for orthostatic intolerance which can be reversed with NOS inhibition. This suggests that both pre- and postsynaptic arterial vasoconstriction may be affected by NO [14,38]. Gallegos et al. demonstrated that serum concentrations of soluble tumor necrosis factor (TNF) receptor 1 in subjects with VVS were diminished during syncope with a change of 958 pg/mL (*p* = 0.0001), whereas in healthy patients, this value did not change. Therefore, it can be inferred that reactivity to nitrates increases with the presence of TNF, leading to vasodilation [39].

## 5. Conclusions

In conclusion, these studies had a relatively small sample size, and unlike our study, used exogenous NOS inhibitors, which act indirectly. Our study involved the endogenous NOS inhibitor, ADMA, which has a pivotal role in NOS inhibition.

In conclusion, we investigated the role of ADMA molecules in patients with VVS using a cohort design. Our results indicated that the alterations in plasma ADMA levels might have a role during syncopal episodes in patients with VVS. Nevertheless, further research is needed to clarify the role of ADMA in the pathophysiology of VVS.

### Limitations

The major limitations of our study stemmed from it being a single-center study and the small sample size. Because of the lack of knowledge regarding the diurnal variation of ADMA levels, HUT tests were performed during the morning, and blood samples were collected before the tests. Furthermore, the NO levels were not measured to evaluate ADMA effects.

## Figures and Tables

**Figure 1 medicina-55-00718-f001:**
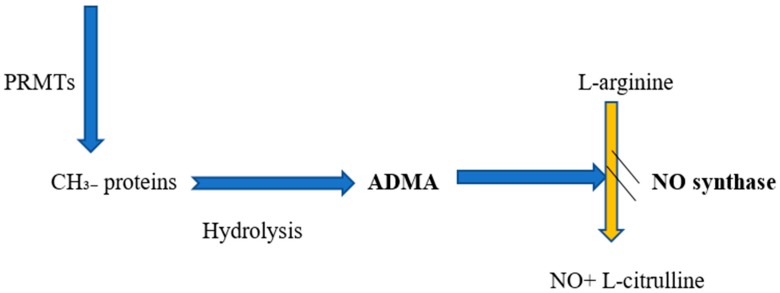
Relationship between ADMA and NO formation. Abbreviations: PRMTs, protein arginine methyltransferases; ADMA, asymmetric dimethylarginine; NO, nitric oxide.

**Figure 2 medicina-55-00718-f002:**
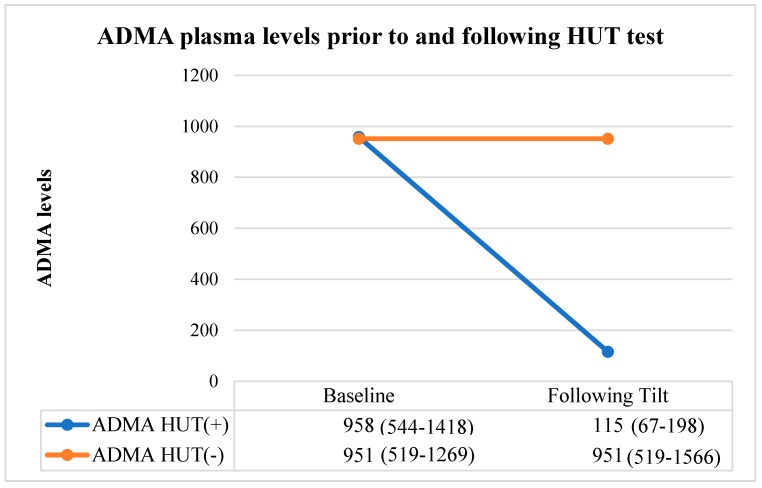
ADMA levels measured prior to and following the HUT test of patients. Abbreviations: HUT, head-up tilt; VVS, vasovagal syncope; ADMA, asymmetric dimethylarginine.

**Table 1 medicina-55-00718-t001:** Demographic and clinical properties between HUT(+) and HUT(−) groups in the study population.

	HUT(+) (*n* = 57)	HUT(−) (*n* = 35)	*p*
Age (year)	24.85 ± 4.01	25.62 ± 3.54	0.354
Gender (women %)	39 (% 68)	21 (% 60)	0.410
LVEF (%)	65.10 ± 5.08	67.00 ± 5.71	0.101
SBP (mmHg)	111.28 ± 11.37	114.57 ± 6.68	0.124
DBP (mmHg)	69.87 ± 8.49	71.28 ± 5.46	0.335
HR (min)	73.56 ± 3.63	74.62 ± 2.98	0.148
TTST (min) VD Type (n,%)	30.22 ± 12.50 30(%53)		
Mixed Type (n,%)	27(%47)		

Abbreviations: LVEF, left ventricular ejection fraction; SBP, systolic blood pressure; DBP, diastolic blood pressure; HR, heart rate; TTST, tilt test syncope time; VD, vasodepressor.

**Table 2 medicina-55-00718-t002:** ADMA levels of HUT (+) and HUT (−) patients.

	All Patients (*n* = 92)	HUT(+) (*n* = 57)	HUT(−) (*n* = 35)
ADMA, ng/mL (before HUT test)	955(543–1269)	958(544–1418)	951(519–1269)
ADMA, ng/mL (after HUT test)	198(87–776)	115(67–198)	951(519–1566)

Abbreviations: HUT, head-up tilt; ADMA, asymmetric dimethylarginine.

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
