# Peer review of "Serum Asymmetric Dimethylarginine Levels in Patients with Vasovagal Syncope"

_medicina, 2019, doi:10.3390/medicina55110718_

Round 1
Reviewer 1 Report
The reviewers have addressed my concerns.
No further revisions needed.
Author Response
Dear Editor,
We would like to thank the reviewers for their comments which helped us to improve the scientific value of the manuscript. The changes made are listed in the following section.
Reviewer #1
This article describes a study of ADMA levels in vasovagal syncope during the head-up tilt test. Some major changes and issues should improve the paper:
Q1.- During all the paper there are some words written together (even in the title), sometimes even three words together. No spaces after ; or : or. This makes the paper difficult to read.
The English should be revised.
A1.This statement has been corrected.English certificate was added.
Q2. - Page 1: A figure for NO pathway should be added to explain the relationship between ADMA and NO formation.
A2. This statement has been corrected.A figure for NO pathway were added as Figure 1.
Q3.- Page 2, line 2: “There are two major inhibitors of NOS…”. However, the authors explain only ADMA. Do you think of SDMA? Which other inhibitor? If ou refer to SDMA, why it is not measured?
A3. ADMA is a member of the methylarginin group. Methylarginins have three forms. 1- ADMA, 2- SDMA, 3- L-NMMA. ADMA and L-NMMA are only inhibitor of nitric oxide synthase. ADMA is major inhibitor of nitric oxide synthase, therefore only ADMA levels were measured.
Q4.- Page 2, line 55: It is not clear if head-up tilt (HUT) test and tilt-table test is the exactly the same test.
A4.This statement has been corrected.Head-up tilt (HUT) test and tilt-table test are the exactly the same test. All tilt-table words changed as Head-up tilt (HUT).
Q5.- Page 2, line 58: 97 patients in Methods are not equal to the sum of 57(+) + 35(-) of the Results. Please, correct or explain which is the difference.
A5.This statement has been corrected. 5 patients who were thought to be conversion were excluded from the study.
Q6.- Page 2, line 78 and line 88: At the beginning of the test it is not clear if the blood is collected, then the patients wait for 15 minutes before the start of the test.
A6. This statement has been corrected.The patients were kept on tilt table in supine position for 15 minutes after blood collection.
Q7.- Page 3, line 123: If ADMA decreased after HUT test, and it is the only positive results, it should be better explained.
A7. This statement has been corrected.ADMA levels decreased in HUT test positive group compared to the before test, whereas ADMA levels did not change in HUT test negative group.
Q8.After test, NO increased and BP decreased. However, NO is not measured and BP does not have significant result before and after test for HUT(+) patients.
A8. Thank you for mentioning this point.As amajor limitation of this study are provided in NO levels are not measured.Blood pressure measurements were not mentioned in the HUT positive group because of low blood pressure.
Q9.- Page 4: Table 2 should be added showing the same parameters as Table 1 but before and after test, for HUT(+), HUT(-) and all patients.
A9. This statement has been corrected.Table 2 added.
Q10.- Page 4, line 161-162: This sentence is not followed by the results of the paper, because blood pressure did not show significant difference.
A10. This sentence is not related to the results of our study and is presented as a general information on this subject.
Q11.- Page 5, line 170: explain more about this sentence. Why and how it is believed that NOS inhibitors play a role in vasovagal syncope?
A11. The role of NO in patients with vasovagal syncope is not well established.
Some experimental data suggest that NO might play a role in the vasodilation occurring during HUT:
--- urinary cGMP has been found to be very high in patients after an HUT- induced syncope when compared with patients with negative responses(1).
--- it has been reported that both the endothelium-dependent and -independent vasodilation is significantly higher in patients with vasovagal syncope than in healthy controls(2).
---Patients with clinical episodes of vasovagal syncope are particularly sensitive to the administration of organic nitrates such as nitroglycerin or isosorbide dinitrate(3,4).
--- Shi et al . reported an increment in plasma levels of NO during HUT in children with syncope and positive tests(5).
---Stewart et al. measured the effect on NO through an inhibitor of NO synthetase (Nc-monomethyl-L-arginine). The blockade reversed all changes observed during the orthostatic challenge in patients with VVS. Their data demonstrate that impaired splanchnic adrenergic vasoconstriction in young patients with VVS, the most common form of syncope, accounts for orthostatic intolerance which can be reversed by NOS inhibition. They conclude that arterial vasoconstriction is impaired in young VVS patients, which is corrected by NOS inhibition. This suggests that both pre- and post-synaptic arterial vasoconstriction may be affected by NO(6).
---Gallegoset al. demonstrated that serum concentrations of soluble TNF receptor 1 in subjects with VVS were diminished at the moment of the syncope with a change of 958pg/mL (P=0.0001) whereas in healthy patients this value did not change. It can be inferred that reac- tivity to nitrates increases with the pres- ence of TNF leading to vasodilation(7).
1.Kaufmann H. Neurally mediated syncope: pathogenesis, diagnosis, and treatment, Neurology, 1995, vol. 45 (pg. S12-8).
Takase B, Akima T, Uehata A, Katushika S, Isojima K, Satomura K, et al. Endothelial function and peripheral vasomotion in the brachial artery in neurally mediated syncope, Clin Cardiol , 2000, vol. 23 (pg. 820-4).3.Raviele A, Gasparini G, Di Pede F, Menozzi C, Brignole M, Dinelli M, et al. Nitroglycerin infusion during upright tilt: a new test for the diagnosis of vasovagal syncope, Am Heart J , 1994, vol. 127 (pg. 103-11).
4.Aerts AJ, Vandergoten P, Dassen WR, Dendale P. Nitrate-stimulated tilt testing enhances the predictive value of the tilt test on the risk of recurrence in patients with suspected vasovagal syncope, Acta Cardiol , 2005, vol. 60 (pg. 15-20).
5.Shi Y, Tian H, Gui YH, He L. Association of nitric oxide and eNOS with the pathogenesis of vasovagal syncope, Zhongguo Dang Dai Er Ke Za Zhi , 2008, vol. 10 (pg. 478-80)
6.stewart JM, sutton r, Kothari ML, et al. nitric oxide synthase inhibition restores orthostatic tolerancein young vasovagal syncope patients. Heart 2017;103:1711–8.
7.Gallegos A, Márquez-Velasco r, Allende r, et al. serum concentrations of nitric oxide and soluble tumor necrosis factor receptor 1 (sTnFr1) in vasovagalsyncope: effect of orthostatic challenge. Int J Cardiol 2013;167:2321–2.
Q12.- Please, check and discuss new articles about ADMA, vasovagal syncope and HUT.
https://www.ncbi.nlm.nih.gov/pubmed/29650798
https://www.ncbi.nlm.nih.gov/pubmed/28501796
A12. This statement has been corrected.Additions from your recommended articles.
Best regards.

Reviewer 2 Report
The authors have revised the manuscript appropriately and upgraded it.
However, there are some reference errors in the new manuscript.
Page 3, line 96: references [24, 25] are correct? It seems to be ref. [23]. Page 3, line 103: ref. [26] may be mistake for ref. [23]? Page 5, line 196-197 and 197-201: need references, may be [6] and [39], respectively.Author Response
Dear Editor,
We would like to thank the reviewers for their comments which helped us to improve the scientific value of the manuscript. The changes made are listed in the following section.
Reviewer #2
The authors demonstrated that ADMA levels significantly decreased just after a positive HUT testing in patients with vasovagal syncope (VVS). This result seems to be supported the theory that NO has a major role in the underlying pathophysiologic mechanism of VVS. The manuscript is well written and very impressive, and valuable for publication. However, there is some concern for acceptance.
Q1.The numbers of references are incorrect. The authors should ascertain the number of references and correct them, especially after 23. These mistakes confuse readers. In addition, the article of reference 5 seems to be not suitable for the international journal. Is there any English title or abstract? The manuscript needs “space” in many sections such as “andgroup2”(page 1, line 18), “andhas”(page 1, line 34), and so on. Please insert “space” in appropriate spaces. The authors need to show the SD range of each ADMA value in figure 1. Page 5, line 171-2: need the reference.
A1.This statement has been corrected.The numbers of references are corrected. Reference 5 is changed as a new reference. SD range of each ADMA value was added on the figure 2.
Best regards

Reviewer 3 Report
This article describes a study of ADMA levels in vasovagal syncope during the head-up tilt test. Major changes have been corrected.
- Page 2: The quality of figure 1 is a bit poor.
Author Response
Dear Editor,
We would like to thank the reviewers for their comments which helped us to improve the scientific value of the manuscript. The changes made are listed in the following section.
Reviewer #3
This is interesting study and supports prior findings by Atici A that ADMA are decreased right after VVS.
Methods and conclusions are appropriate.
Q1.How did the authors define positive HUTT? Did ADMA decrease if patients had pre-syncope rather than syncope? Was there any difference depends on the type of VASIS classifications?
A1.This statement has been corrected.
Bradycardia was defined as a 20% decrease from the baseline HR or a decrease of more than 20 beats in a minute. Hypotension was defined as a decrease of systolic blood pressure (SBP) more than 20% of baseline or maximum blood pressure or more than 30 mmHg decrease of systolic pressure.
Syncope occurred in all HUT positive patients during the test. therefore, it is not known whether the decrease in ADMA levels is associated with presyncope.
To compare the subgroup of HUT(+) patients, and no significant difference was detected following the HUT test(ADMA of 123 [68–223] in the vasodepressor type VVS vs. 99 [60–164] in the mixed type VVS; p = 0.187).there is no difference depends on the type of VASIS classifications.
Best Regards
